# What Are We Detecting, Really? LLM-Generated Text Detection Remains an Unsolved Problem

## Abstract

This position paper argues that, in most practical cases, it is not possible to accurately detect LLM-generated text. We consider that "LLM-generated text" refers to the content produced by LLMs through normal prompts. As implied by the names "LLM-generated text" and "human-written text", the difference lies in how they are produced, but in practice, we can only evaluate them based on the final output—the text itself—where there is often significant overlap between human- and machine-generated content.. The numerical results of LLM-generated text detection are often misunderstood and their significance is diminishing. The detectors can serve a purpose under specific conditions, whose results should only be used as a reference with greater caution rather than the decisive indicator.

## 1 Introduction

The rapid development of large language models (LLMs) has led to a rise of LLM-generated text, which has been observed across various fields, such as academia [Liang et al., 2024, Geng and Trotta, 2024], Wikipedia [Brooks et al., 2024, Huang et al., 2025], and numerous online texts [Sun et al., 2024, Liang et al., 2025]. The detection of LLM-generated text has attracted the attention of researchers, and many detectors have been proposed and studied [Yang et al., 2023, Wu et al., 2025]. Before starting our discussion, we want to clarify the following definition:

*What exactly is "LLM-generated text"?*

In fact, the term "LLM-generated text" is fairly new. Researchers also used expressions like "machine-generated text" or "AI-generated", as seen in Table 1. For simplicity, we use "LLM-generated text" to refer to the subject of study in this paper, as it is more precise than the other expressions. Apart from slight differences in terminology, the definition of "LLM-generated text" in most papers is quite broad, meaning the text can be produced in many ways using LLMs, like paraphrasing, translation, or generating long text from simple prompts. Therefore, we can consider that **"LLM-generated text" refers to the content produced by LLMs through normal prompts**.

The reliability of the detectors has also been widely discussed [Sadasivan et al., 2023, Chakraborty et al., 2024]. The indistinguishability between LLM-generated and human-written text is one of big challenges for LLMs [Kaddour et al., 2023]. Similar to the central question in Chakraborty et al. [2024]'s work, we explore the following key one:

*Is it possible to detect the LLM-generated text in practice?*

Literally speaking, LLM-generated text detectors need to cover all these different scenarios, but few detectors have tried to distinguish them [Cheng et al., 2025]. If we take into account the diversity of LLMs and prompts, as well as human-in-the-loop, the situation becomes even more intricate. Hence, we argue that, **in most practical cases, it is not possible to accurately detect LLM-generated text.**

Table 1: Definition of LLM-generated text in different papers

| Paper | Definition |
|---|---|
| Crothers et al. [2023] | *"Machine-generated text" is natural language text that is produced, modified, or extended by a machine.* |
| Kumarage et al. [2024] | *In this survey, we define AI-generated text as output produced by a natural language generation pipeline employing a neural probabilistic language model.* |
| Wu et al. [2025] | *LLM-generated Text is defined as cohesive, grammatically sound, and pertinent content generated by LLMs.* |

While limitations of these detection methods have caused concern among researchers [Sadasivan et al., 2023, Liang et al., 2023, Doughman et al., 2024, Nicks et al., 2023, Saha and Feizi, 2025], they could be applied in diverse contexts. The emergence of the GPTZero platform is a good example, although it is currently unclear how frequently people use it. So the next question arises naturally:

*Should we use these detectors?*

We think that **the detectors can serve a purpose under specific conditions, whose results should only be used as a reference with greater caution rather than the decisive indicator.**

We will discuss them in detail in the following sections.

## 2 Detectors

Probably most people became aware of LLMs after the release of ChatGPT, but the research on detecting text generated by language models had started before that. For example, Gehrmann et al. [2019] proposed the GLTR tool to detect whether text was generated by models, with experiments involving GPT-2 [Radford et al., 2019] and BERT [Devlin et al., 2019]. Zellers et al. [2019] developed the Grover model to detect AI-generated fake news. Even GPT-3 [Brown et al., 2020] continued to face skepticism regarding its text-generation capabilities [Bender et al., 2021], making the detectors' performance unsurprising.

Another pioneering work by [Ippolito et al., 2019] demonstrated that humans have already encountered some difficulties in identifying texts generated by GPT-2. Later, Clark et al. [2021] found that untrained people at the time were not very good at recognizing text produced by GPT-3, and Wahle et al. [2022] noticed the similar situation for machine-paraphrased plagiarism.

In the past two or three years, the rapid development and spread of LLMs has drawn significant attention from researchers to the detection of LLM-generated text, and diverse methods have been proposed [Wu et al., 2025]: DetectGPT [Mitchell et al., 2023], Fast-DetectGPT [Bao et al., 2023], DetectLLM [Su et al., 2023], LLMDet [Wu et al., 2023], DeID-GPT [Liu et al., 2023] and some others [Dugan et al., 2023] in 2023; Binoculars [Hans et al., 2024], TOCSIN [Ma and Wang, 2024], Dpic [Yu et al., 2024b], Text Fluoroscopy [Yu et al., 2024a] in 2024. The examples listed above are illustrative, and the actual number of detectors is much larger.

In the meantime, specialized detectors have been developed, for instance, targeting journalistic news articles [Bhattacharjee et al., 2023] and tweets [Kumarage et al., 2023, Gambini et al., 2022]. Additionally, the detection of LLM-generated text is not limited to English [Wang et al., 2025]. we have also seen detectors for other languages, such as French [Antoun et al., 2023a], Japanese [Zaitsu and Jin, 2023], Chinese [Wang et al., 2024a].

While these techniques of detection performed well earlier on certain datasets, the ongoing progress of LLMs also makes detection harder [Wu et al., 2025]. A wide range of methods are utilized by these detectors, but the absence of universal benchmarks and different application scenarios limit meaningful comparison. We will address this issue in greater detail in Section 5.1.

These detection methods can be classified into many categories according to different criteria. For instance, Abdali et al. [2024] classifies them as supervised methods, zero-shot methods, retrieval-based methods, watermarking methods, discriminating features. Wu et al. [2025] mainly examines

them through the lens of watermarking techniques, statistics-based detectors, neural-based detectors, and human-assisted methods. It is difficult to provide a comprehensive summary of LLM-generated text detectors, but to our knowledge, no detector has been conclusively established as the best, particularly in practical deployment contexts.

There are other ways to categorize the detectors. For example, most studies only think about binary classification, and detectors with multi-category cases have rarely been explored, which will be further examined in Section 5.4.

# 3   Related Work

**Benchmark**   One of the major challenges in establishing benchmarks for detecting LLM-generated text is that LLMs are continuously evolving, and their characteristics do not remain the same. For example, [Liyanage et al., 2022] created their benchmark with GPT-2, which should be quite differently from the current advanced LLMs. More LLMs were employed in subsequent benchmark construction [Wang et al., 2024b, He et al., 2024, Cornelius et al., 2024], but the number of prompts and scenarios used was limited. Some recent benchmarks [Tao et al., 2024, Wu et al., 2024] have incorporated a broader range of scenarios, and their impact and effectiveness remain to be seen. Similar challenge for the dataset [Gritsai et al., 2024].

**Watermarking and Attack**   As mentioned earlier, watermarking LLMs is considered a category of detection methods, and it has shown good effectiveness in simulation [Kirchenbauer et al., 2023], which may also be an ethical necessity [Grinbaum and Adomaitis, 2022]. Researchers have proposed diverse watermarking techniques [Jovanović et al., 2024, Dathathri et al., 2024, Christ et al., 2024], but given the variety of attack techniques, this game is far from over [Krishna et al., 2023, Cai and Cui, 2023, Sadasivan et al., 2023, Lau and Zubiaga, 2024, Tufts et al., 2024, Macko et al., 2024].

**Evaluation and Test**   These different methods have also been tested by various researchers [Weber-Wulff et al., 2023, Walters, 2023], in the beginning, some issues have been pointed out, such as biases [Liang et al., 2023] and false positives [Rashidi et al., 2023, Giray, 2024]. Although different detectors have been proposed over time, many challenges still exist [Kar et al., 2024, Tufts et al., 2024]. Given the diversity of usage scenarios and the variety of LLMs, such results are to be expected. We will provide a detailed analysis later.

# 4   Case Study

For a simple illustration of LLM-generated text and detection, we take the introduction from the website of NeurIPS[1] for simulations. This paragraph is processed with various prompts across different LLMs (DeepSeek-V3 [Liu et al., 2024], DeepSeek-R1 [Guo et al., 2025], GPT-3.5, GPT-4o-mini, GPT4o [Hurst et al., 2024]).

We present the original text and two generated texts with different prompts, and more examples are provided in the Appendix A.

**Original Text**   *The conference was founded in 1987 and is now a multi-track interdisciplinary annual meeting that includes invited talks, demonstrations, symposia, and oral and poster presentations of refereed papers. Along with the conference is a professional exposition focusing on machine learning in practice, a series of tutorials, and topical workshops that provide a less formal setting for the exchange of ideas.*

**DeepSeek-V3**   ("Polish the following passage: . . . ") *Founded in 1987, the conference has grown into a premier multi-track interdisciplinary annual event. It features invited talks, demonstrations, symposia, and peer-reviewed oral and poster presentations. Alongside the conference, attendees can explore a professional exposition highlighting practical applications of machine learning, a series of tutorials, and topical workshops that foster informal idea exchange.*

---

[1] https://neurips.cc/

**DeepSeek-R1**  ("Polish the following passage: . . .") *Established in 1987, the conference has grown into a premier interdisciplinary event held annually, featuring multiple tracks of programming. Attendees engage with invited talks, demonstrations, symposia, and peer-reviewed research presented through both oral and poster sessions. Complementing the core program are a professional exposition showcasing real-world applications of machine learning, hands-on tutorials, and topical workshops designed to foster informal dialogue and collaborative idea exchange.*

**GPT-4o**  ("Polish the following passage: . . .") *Established in 1987, the conference has evolved into a comprehensive, multi-track interdisciplinary annual event. It features invited talks, demonstrations, symposia, and both oral and poster presentations of peer-reviewed papers. In addition to the main conference, there is a professional exposition dedicated to practical applications of machine learning, as well as a series of tutorials and topical workshops. These workshops offer a more informal setting for the exchange of ideas.*

**GPT-4o**  ("Rewrite the following passage without using complex expressions: . . .") *The conference started in 1987 and now happens every year with multiple sessions. It features invited talks, demonstrations, discussions, and presentations of selected papers. There is also a professional expo about using machine learning, a series of tutorials, and workshops that offer a more relaxed space for sharing ideas.*

Although they modify the original text (for example, the underlined words above), the added words and expressions are not the same. Table 2 presents the results of detecting these texts using Fast-DetectGPT [Bao et al., 2023]. Even though these texts are all generated by LLMs, their detection outcomes vary widely. As we can easily find, in this case, the detector struggles to clearly identify text generated by DeepSeek-V3 and DeepSeek-R1, as the probability of texts by them being identified as machine-generated is even lower than that of the original text from NeurIPS website. The results of GPT-3.5, GPT-4o-mini, and GPT-4o show that prompts can easily affect the ouputs and the detection results.

Table 2: Detection results using Fast-DetectGPT. The last two columns correspond to the predictions of the machine-generated results when the Sampling/scoring model is gpt-neo-2.7b and falcon-7b, respectively.

| Prompts | Model | p1 | p2 |
|---|---|---|---|
| (original text) | - | 44% | 23% |
| Polish the following passage: | DeepSeek-V3 | 34% | 12% |
|  | DeepSeek-R1 | **21%** | **10%** |
| Polish the following passage: | GPT-3.5 | 50% | 22% |
|  | GPT-4o-mini | 35% | 20% |
|  | GPT-4o | **84%** | **75%** |
| Rewrite the following passage without using complex expressions: | GPT-3.5 | 30 % | 16 % |
|  | GPT-4o-mini | 51% | 55% |
|  | GPT-4o | 38% | 15% |

These are merely a few basic examples of the issues and limitations faced by LLM-generated text detectors. A more detailed discussion will follow in the next section.

## 5   Issues and Limitations

As we briefly introduced before, the detection of text generated by LLMs has emerged as a widely discussed and actively pursued task in natural language processing. Such detection tools are often promoted for their potential utility in identifying instances of plagiarism [Pudasaini et al., 2024], academic dishonesty (e.g., cheating during examinations) [Wang and Li, 2025], the automatic generation of unethical peer reviews [Kumar et al., 2025], and other forms of content manipulation.

Meanwhile, many issues and challenges have been discussed [Tang et al., 2024, Wu et al., 2025, Fraser et al., 2024, Abdali et al., 2024]. Despite their apparent usefulness, there are some fundamental

limitations associated with these tools that raise serious ethical and methodological concerns. We will address these issues and limitations from various perspectives in this section.

## 5.1 Lack of Precise Definition and Gold-Standard Benchmark

Unlike most question-answering or classification tasks, "human-written text" itself lacks a clear and well-defined boundary compared to "LLM-generated text". **As implied by their names, the difference lies in how they are produced, but in practice, we can only assess them based on their final output i.e., the text, where in which a lot of overlap between them.**

Researchers often say that the text generated by LLMs is different from that written by humans [Muñoz-Ortiz et al., 2024, Reinhart et al., 2025]. Just as different people can write in different styles, LLMs can also generate varied outputs. We think that what is commonly referred to as "LLM-generated text" is only a subset of the text that LLMs can potentially produce, and it's likely the kind that corresponds to the most common and direct prompts. For instance, many detectors are trained on text generated by LLMs, which cannot represent all possibilities. Consequently, their detection capabilities are constrained. While different parameters can be set for various types of cases [Hans et al., 2024], such configurations can hardly cover all possible scenarios.

As we mentioned in related work, although some researchers have proposed benchmarks for detecting LLM-generated text [Liyanage et al., 2022, Wang et al., 2024b, He et al., 2024, Tao et al., 2024, Wu et al., 2024], their adoption has not yet become as widespread as other well-known LLM benchmarks, such as GLUE [Wang et al., 2018] and MMLU [Hendrycks et al., 2020]. Although these benchmarks have also faced some criticism [Hadi et al., 2023], there is still no highly universal benchmark for detecting LLM-generated texts.

Besides, due to the diversity of usage scenarios and the continuous updates of LLMs, a gold-standard benchmark is hard to realize, may even remain permanently absent.

## 5.2 Inherent Imperfection of Detection Tools

No existing LLM-detection system is infallible. In real-world conditions, a detection accuracy of 85% is typically considered outstanding. Yet, this figure necessarily implies a 15% error rate, which may include both false positives and false negatives.

False positives—in which human-written content is incorrectly flagged as machine-generated—are particularly problematic in high-stakes contexts such as academic integrity investigations. And the problem of false positives has already been observed and discussed. For example, Rashidi et al. [2023] found that the AI text detector erroneously identified up to 8% of the known real abstracts as AI-generated text, and Giray [2024] states that false positives disproportionately affect non-native English speakers and scholars with distinctive writing styles. In addition, Tufts et al. [2024] think that adversarial attacks can easily bypass these detectors, and balancing high sensitivity with a reasonable true positive rate remains challenging.

Accusing someone of misconduct based on an imperfect tool can lead to unjust outcomes, reputational damage, and institutional distrust. Therefore, even detectors with relatively high accuracy present significant risks when used for evaluative or disciplinary purposes.

Experiments also show that certain detectors may exhibit bias against non-native English writers [Liang et al., 2023] or against certain demographic groups [Kadoma et al., 2025]. The analyses from Li and Wan [2025] revealed that all the detectors they tested are highly sensitive to CEFR level and language environment. With LLMs being so widely used in academia [Eger et al., 2025, Russell et al., 2025], detecting AI-generated text must be handled with extreme care.

These detectors also face numerous other challenges, including difficulties in short-text detection [Gameiro et al., 2024, Shi et al., 2024] and the issues of modification and classification that we will discuss later. As such, current detectors are far from perfect and may never achieve perfection in the future either.

## 5.3 Poor Robustness to Textual Modifications

There have always been many doubts about the effectiveness of these detectors [Sadasivan et al., 2023, Weber-Wulff et al., 2023]. Another issue pertains to the brittleness of these tools in realistic scenarios. An early study has shown that while humans can reliably detect poetry produced by GPT-2, but they struggle to accurately recognize it after human selection [Köbis and Mossink, 2021]. If post-generation modifications are taken into account, the detection process should become more challenging.

Most current detectors are trained to recognize text that has been directly generated by an LLM without post-editing. While some recent systems claim to maintain performance when the LLM-generated text is lightly modified, empirical evidence shows that detection accuracy tends to decline as the extent of human revision increases.

In practice, LLM-generated content is often edited, paraphrased, or interwoven with human-written material, especially in academic contexts. Consequently, the tools' applicability to real-world use cases remains limited. This limitation exacerbates the concerns raised in the first point, as reliance on imperfect systems in nuanced or ambiguous situations increases the likelihood of erroneous judgments.

## 5.4 A Wide Variety of Use Cases and the Limits of Binary Classification

A fourth, and perhaps more fundamental, concern lies in the heterogeneity of LLM-generated content.

The ethical implications of LLM use depend heavily on the context and intent of usage. For instance, a researcher who uses LLMs to generate entire manuscripts with minimal intellectual input contributes to the proliferation of unoriginal work, thereby burdening peer-review systems and undermining the credibility of scholarly communication.

Such practices are clearly unethical. In contrast, a non-native speaker might use an LLM to translate, rephrase, or refine a manuscript that is otherwise the product of original research. In this case, the LLM acts as a language aid rather than a generator of substantive content. Yet most detection systems treat these qualitatively different scenarios in the same manner, reducing the complex spectrum of authorship to a binary classification of "human-written" versus "machine-generated". Similar problems have also been noted in very recent studies [Lepp and Smith, 2025].

Generally, most studies focus on the binary classification problem of determining whether a given text is generated by LLMs. While some detection methods could achieve good results on given datasets, the scenario becomes more much complicated in real-world settings. For example, people could edit LLM-generated text or mix it with human written text, which has also attracted considerable attention [Zhang et al., 2024a, Abassy et al., 2024, Kumar et al., 2025]. Only a small number of researchers have tried to identify specific roles of LLM in content generation [Cheng et al., 2025], and no universally accepted approaches have been established.

## 5.5 Diversity in LLMs

Even without considering the usage scenarios noted before, different LLMs generate text in different styles [Rosenfeld and Lazebnik, 2024, Sun et al., 2025]. Empirical studies have consistently demonstrated that different LLMs exhibit distinct stylistic patterns fingerprints, which could even be used for classification [McGovern et al., 2024, Sun et al., 2025, Bitton et al., 2025].

Studies indicate that the detectability of texts depends on the LLM used for text generation [Antoun et al., 2023b]. For example, Wu et al. [2024] pointed out that the Binoculars [Hans et al., 2024] only achieved a 55.15% AUROC in detecting texts generated by Claude, while for texts generated by several other models, it reached at least 88%. A comparable point is reflected in Table 2.

Detectors may more easily flag text from older and smaller models [Elkhatat et al., 2023, Saha and Feizi, 2025]. As we all know, the development of LLMs has not stopped, so the timeliness of detectors is also another challenge. Obviously, the same LLM can produce different texts in response to different prompts for the same task, as we have shown before. **Although these detectors may still be applicable in certain scenarios, their use requires greater caution.**

## 5.6 Others

Those familiar with LLMs and detectors are aware of the potential issues, but the public tends to be easily drawn to these numbers and brief conclusion. The lack of detector interpretability represents another concern [Ji et al., 2024], severely limiting the ability to provide transparent explanations to the public.

The appropriate use of LLMs has now been widely accepted, such as in the NeurIPS submission process [2]. In addition to the examples given earlier, the traces of LLM-generated text have now been found in various fields, such as student essays' answers [Leppänen et al., 2025] and words used in speaking [Yakura et al., 2024, Geng et al., 2024].

The abuse and misuse of these detectors can create ethical risks. Meanwhile, the numerical effectiveness of LLM-generated text detectors is declining. On the one hand, human may be influenced by LLMs and may create text resembling LLM-generated text. One the other hand, people may also adapt their language to bypass LLM detection tools [Geng and Trotta, 2025].

Therefore, **when interpreting the detection results of LLM-generated text, it is necessary to explicitly specify which kind of subset serves as the reference to establish the detector**.

# 6 Positive Impact of LLM Usage

The social impact of of LLMs has already been considered [Solaiman et al., 2019].

While LLM-generated text is frequently the subject of criticism—particularly due to concerns around academic dishonesty, plagiarism, and fraud, which have led to the development of various detection tools—it is equally important to emphasize the legitimate and ethical uses of large language models. As discussed earlier, LLMs can play a valuable role in numerous contexts. For instance, they help bridge linguistic divides by enabling non-native speakers to produce coherent and idiomatic texts in English or other target languages, thereby supporting greater inclusivity in academic and professional communication. They also facilitate high-quality machine translation, making content in multiple languages more accessible, and allow for the efficient synthesis of large textual corpora, which can aid research and knowledge production.

Rather than focusing solely on the detection and policing of LLM-generated text, it may be more productive to advocate for transparency regarding their use. In academic publishing, for example, it is increasingly common to disclose how LLMs have assisted in drafting, editing, or rephrasing portions of a manuscript.

Such uses are generally limited to improving expression or exploring alternative formulations; the substantive intellectual work remains the responsibility of human authors. Importantly, LLMs should not be considered co-authors nor used to autonomously generate scientific content in its entirety. Clear guidelines and disclosures can thus help normalize the ethical integration of LLMs into scholarly workflows without undermining academic integrity.

People began discussing ChatGPT's positive impact shortly after its emergence [Kasneci et al., 2023]. Non-native English speakers have to put in more effort as scientists, and there has been discrimination in the past [Amano et al., 2023, Lepp and Smith, 2025]. Automatic editing methods have shown promise in improving alignment between LLM-generated and human-written text [Chakrabarty et al., 2024]. If LLMs are applied properly and people assess detection tools reasonably, their positive influence can be greatly amplified.

# 7 Alternative Views

Researchers have not yet reached full agreement on the detectability of LLM-generated text.

Chakraborty et al. [2024] claim in their position paper: "Despite ongoing debate about the feasibility of such differentiation, we present evidence supporting its consistent achievability, except when human and machine text distributions are indistinguishable across their entire support. Drawing from

---

[2]`https://neurips.cc/Conferences/2025/LLM`

information theory, we argue that as machine-generated text approximates human-like quality, the sample size needed for detection increases."

But we believe that LLMs are fully capable of generating text that is nearly indistinguishable from human-written content. Furthermore, practical observations have shown that humans possess the capacity to identify LLM-generated text with reasonable accuracy [Russell et al., 2025], and such coevolution may already be occurring [Geng and Trotta, 2025]. These challenges in these real-world data cannot be resolved by increasing the sample size.

The key disagreement among researchers may not be technical in nature, but rather stems from differing perspectives on human intervention. Take watermarking studies as an example, if people edit the generated text (which is simple to do), the watermark's reliability may be greatly weakened [Dathathri et al., 2024].

There are also researchers who share similar views with us. For example, Zhang et al. [2024b] argue that "We believe that the issue of AI-generated text detection remains an unresolved challenge. As LLMs become increasingly powerful and humans become more proficient in using them, it is even less likely to detect AI text in the future." And Nicks et al. [2023] "advise against continued reliance on LLM-generated text detectors".

# 8 Future Perspectives and Predictions

LLMs were compared to stochastic parrot [Bender et al., 2021] a couple of years ago, but their capabilities are gradually being recognized [Srivastava et al., 2022], and now their competencies have reached or even surpassed those of human experts in various fields. This leads us to speculate that future LLM-generated texts could surpass current versions in human resemblance. The persona effect could be a good example [Hu and Collier, 2024].

The gap between LLM-generated and human-written text is expected to narrow further. As noted by several researchers, humans can learn to detect AI-generated texts [Milička et al., 2025] and can become accurate and robust detector of LLM-generated text [Russell et al., 2025].

And therefore, we can find more cases of coevolution [Pedreschi et al., 2024, Geng and Trotta, 2025] between human and Human in the future. Consequently, in the future, detecting LLM-generated text may become less important, particularly in terms of numerical interpretation.

Accounting for model collapse [Shumailov et al., 2024, Guo et al., 2023] and knowledge collapse [Peterson, 2025], the detection results may become even more intriguing. Detection efforts should target substantive content (e.g., fact-checking) rather than linguistic characteristics [Schuster et al., 2020].

# 9 Conclusions

Given the current state-of-the-art, existing tools are not equipped to make such fine-grained distinctions. They are structurally unable to assess the proportion, function, or ethical significance of LLM contributions in a given text. As a result, the development and deployment of LLM-generated text detectors raise serious concerns, not only due to technical limitations but also because they risk enforcing overly simplistic and potentially unjust frameworks for evaluating authorship and intellectual responsibility.

And some of these difficulties are simply unavoidable, not merely temporary technical challenges. Texts generated by LLMs and those generated by humans often overlap greatly, with no obvious separation. Moreover, as LLMs become more widely used and people may learn from their outputs, the difference between them may get further smaller. Since text is different from images, it is difficult to balance both the watermark and the original textual information.

While acknowledging that detectors for LLM-generated text can serve a purpose in certain scenarios, we recommend using them with greater caution. And the detection results should only be used as a reference rather than the decisive indicator.

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

# A Simulation Results

- **DeepSeek-V3 (Polish the following passage:).** Founded in 1987, the conference has grown into a premier multi-track interdisciplinary annual event. It features invited talks, demonstrations, symposia, and peer-reviewed oral and poster presentations. Alongside the conference, attendees can explore a professional exposition highlighting practical applications of machine learning, a series of tutorials, and topical workshops that foster informal idea exchange.

- **DeepSeek-R1 (Polish the following passage:).** Established in 1987, the conference has grown into a premier interdisciplinary event held annually, featuring multiple tracks of programming. Attendees engage with invited talks, demonstrations, symposia, and peer-reviewed research presented through both oral and poster sessions. Complementing the core program are a professional exposition showcasing real-world applications of machine learning, hands-on tutorials, and topical workshops designed to foster informal dialogue and collaborative idea exchange.

- **GPT-3.5 (Polish the following passage:).** Established in 1987, the conference has evolved into a multi-track interdisciplinary annual event featuring invited talks, demonstrations, symposia, and oral and poster presentations of refereed papers. In addition, there is a professional exposition dedicated to machine learning in practice, a variety of tutorials, and topical workshops that offer a more relaxed environment for the sharing of ideas.

- **GPT-4o-mini (Polish the following passage:).** Founded in 1987, the conference has evolved into a dynamic multi-track interdisciplinary annual event featuring invited talks, demonstrations, symposia, and both oral and poster presentations of refereed papers. Complementing the conference is a professional exposition dedicated to practical applications of machine learning, as well as a series of tutorials and topical workshops that create a more informal atmosphere for exchanging ideas.

- **GPT-4o (Polish the following passage:).** Established in 1987, the conference has evolved into a comprehensive, multi-track interdisciplinary annual event. It features invited talks, demonstrations, symposia, and both oral and poster presentations of peer-reviewed papers. In addition to the main conference, there is a professional exposition dedicated to practical applications of machine learning, as well as a series of tutorials and topical workshops. These workshops offer a more informal setting for the exchange of ideas.

- **GPT-3.5 (Rewrite the following passage without using complex expressions:).** The conference started in 1987 and has grown into a yearly event that covers different subjects and includes talks, demonstrations, symposiums, and presentations of reviewed papers. There is also an exhibition on machine learning, tutorials, and workshops for sharing ideas in a casual setting.

- **GPT-4o-mini (Rewrite the following passage without using complex expressions:).** The conference started in 1987 and is now held every year. It features various sessions that include invited talks, demonstrations, and both oral and poster presentations of reviewed papers. There is also a professional exhibit that highlights real-world applications of machine learning, along with a series of tutorials and workshops that offer a more relaxed atmosphere for sharing ideas.

- **GPT-4o (Rewrite the following passage without using complex expressions:).** The conference started in 1987 and now happens every year with multiple sessions. It features invited talks, demonstrations, discussions, and presentations of selected papers. There is also a professional expo about using machine learning, a series of tutorials, and workshops that offer a more relaxed space for sharing ideas.

