# OpenReview forum: "What Are We Detecting, Really? LLM-Generated Text Detection Remains an Unsolved Problem"
_NeurIPS.cc/2025/Position_Paper_Track — Submitted to NeurIPS 2025 Position Paper Track_

### Official Review · Reviewer_Krpo · 2025-07-23

**Significance:** 3
**Presentation:** 2
**Rating:** 6
**Confidence:** 4

**Summary:**

This position paper argues that reliably detecting LLM-generated text in most practical settings is impossible, due to the fundamental overlap between human- and machine-written outputs, the diversity of LLMs and prompts, and the inherent limitations of existing detection methods. The paper surveys a wide range of detection approaches, benchmarks, and empirical findings, illustrating issues like high false positive rates, poor robustness to editing, and ethical concerns. The authors caution against the use of detection tools and instead advocate for transparency and clear guidelines on the ethical use of LLMs. The paper also contrasts its position with alternative views that are more optimistic about detectability and discusses future directions.

**Strengths:**

1. **Comprehensive coverage of the topic.** The paper does a commendable job of surveying the landscape of LLM-generated text detection. It reviews a wide array of detection methods, including watermarking, supervised and unsupervised approaches, and benchmark datasets, and discusses their evolution over time. The discussion is supported by a rich set of references, covering both technical aspects (e.g., algorithmic performance, robustness issues) and societal concerns (e.g., fairness, ethics, bias).
  2. **Clear and consistent position throughout the paper.** The authors maintain a clear, principled stance — that detection is unreliable and should be used only with caution — and articulate it consistently across all sections without ambiguity.
  3. **Consideration of social and ethical implications.** The paper goes beyond technical issues to highlight how misuse of detection tools can exacerbate discrimination against certain groups, damage trust, and enforce simplistic models of authorship that do not reflect the collaborative and iterative nature of writing. The acknowledgment of these broader implications adds depth to the analysis.

**Weaknesses:**

1. Although the case study (Section 4) is illustrative, it is small and limited to a single paragraph rewritten under several prompts and evaluated by Fast-DetectGPT. The paper does not include more comprehensive experiments or new quantitative analyses that systematically validate its thesis.
  2. Several limitations are raised in multiple sections with similar phrasing. For example, the problems of benchmark inadequacy are discussed both in the introduction, Section 3, and again in Section 5.1, but without much additional insight. This repetition dilutes the focus and makes the paper longer than necessary to convey its core thesis. Also, the six listed limitations are presented as parallel points, but many are logically connected, making the section feel disorganized and repetitive. For example, the lack of clear definitions and benchmarks (5.1) underpins flaws like poor robustness (5.3) and imperfect tools (5.2), yet these are discussed in isolation.
  3. While the paper concludes by recommending “greater caution” and advocating transparency, it does not provide concrete policy recommendations or practical frameworks. For example, are there hybrid human-in-the-loop verification schemes worth pursuing?

**Questions:**

1. As the authors argue, reliable detection of LLM-generated text is infeasible in practice. What alternative mechanisms or policies do they recommend to uphold academic integrity, prevent misconduct, and maintain trust in scholarly communication? For example, should institutions rely more on disclosure guidelines or human verification?

  2. The paper critiques binary classification as overly simplistic, but does not discuss how different contexts (e.g., academic research, journalism, education, casual social media) may warrant different standards or thresholds for acceptable AI involvement. For example, academic settings often demand strict disclosure or prohibition to preserve integrity, journalism emphasizes factual accuracy and source transparency, while casual social media posts tolerate more AI assistance. Could the authors elaborate on how such context-specific norms and expectations could be incorporated into the evaluation and regulation of LLM-generated content?

**Alternative Position:**

Yes, and alternative positions are well-considered and addressed by the argument

**Author Identification:**

No.

**Context:**

3

**Discussion:**

2

**Ethics:**

["NO or VERY MINOR ethics concerns only"]

**Position:**

Yes, the paper argues for or against a position related to machine learning.

**Support:**

2

**Thoroughness:**

3

---

### Official Review · Reviewer_9JVK · 2025-07-31

**Significance:** 3
**Presentation:** 3
**Rating:** 8
**Confidence:** 4

**Summary:**

This position paper argues that current benchmarks for detecting LLM-generated (scientific) content are inadequate and that a more thoughtful framework is needed. The authors categorize detection signals, propose design principles for benchmarks and analyze detector behavior across different conditions. The paper critiques current practices and offers guidelines to improve future benchmarks. It positions itself against binary LLMs vs human detection approach and calls for more principled metrics and definitions.

**Strengths:**

The paper presents a critique of existing LLM detection benchmarks highlight that most current systems fail to consider the nuances of writing, specifically, scientific writing and detection failure modes. The authors present concrete guidelines for what makes for a good benchmark and show trade-offs and risks in the current evaluation pipelines. The position is relevant to the scientific community as LLM usage has been increasing across research and publication work.

**Weaknesses:**

1. The evaluation of p1 and p2 model generations lack sufficient methodological detail to be fully understandable:

- It is unclear what detection tools were used and why only these specific models were analyzed. It feels somewhat arbitrary. Therefore it is unclear how much these findings are reproducible and generalizable these findings are.
- Is one prompt setting and associated example sufficient for simulation results?
- Why are 4 model output examples given in the main body of the paper and then repeated fully again with 2 more at the end of the paper?

2. The paper would benefit from a table or similar overview listing known LLM detection methods, their core detection signals and known weaknesses or failure causes. In other words, a more detailed overview of Section 5.1 motivating lack of gold-standard benchmarks and why would be useful.

3. The paper claims in Section 5.2 that 85% accuracy is typically outstanding, but this appears to be an unsupported assertion.

Overall the analysis blurs the lines between a survey and a position paper. Some empirical results are presented without enough methodological transparency that may need further validation.

**Questions:**

- Why were these specific models chosen? Would it strengthen the paper position to add more examples or models from different domains?
- Has detection performance degraded as newer LLMs have emerged or are detectors equally ineffective across all model generations?
- If 85% is considered as outstanding accuracy, where does this threshold originate? Is it based on empirical evidence or prior literature? A citation or benchmark comparison would help support this statement.
- Would the authors consider sharing prompts and code along with the position paper? This would help reproducibility of the findings.

**Alternative Position:**

Yes, and alternative positions are well-considered and addressed by the argument

**Author Identification:**

No.

**Context:**

3

**Discussion:**

4

**Ethics:**

["NO or VERY MINOR ethics concerns only"]

**Position:**

Yes, the paper argues for or against a position related to machine learning.

**Support:**

3

**Thoroughness:**

4

---

### Official Review · Reviewer_esMk · 2025-08-25

**Significance:** 4
**Presentation:** 3
**Rating:** 6
**Confidence:** 4

**Summary:**

This paper argues that reliably distinguishing LLM generated text from human written text is unsolvable. The authors review existing detectors and benchmarks, highlighting fundamental issues and limitations. These include the lack of a precise definition of LLM generated text, the absence of gold standard or universal benchmarks, and the inherent imperfections of current tools, such as frequent false positives and false negatives. Detection accuracy is also undermined by sensitivity to textual modifications, the oversimplified binary framing of detection tasks (which fails in mixed human AI authored scenarios), and the diversity of outputs across different LLMs. Beyond technical flaws, the paper emphasizes the ethical risks of over relying on imperfect detectors, particularly in high-stakes settings . The authors conclude that while detectors may be useful in limited contexts, their outputs should be treated only as advisory signals references rather than decisive indicators.

**Strengths:**

• The paper addresses the urgent question of whether AI generated text can actually be detected.

• The authors present a small case study and conclude that reliable detection is practically unsolvable, illustrating this through different scenarios that make the debate engaging.

• Key limitations are identified:

	• Lack of a precise definition of LLM generated text.
	• Absence of gold standard or universal benchmarks.
	• Inherent flaws of existing detectors (frequent false positives and false negatives).
	• Sensitivity of detection results to small textual modifications.
	• Oversimplified binary framing, which fails in cases where human and AI text are mixed.
	• Diversity of outputs across different LLMs, which complicates detection.

• The paper also highlights the ethical dimension, showing how unreliable detectors can unfairly penalize non-native speakers and marginalized communities, partly due to limited representation in training data.

• The authors conduct a thorough literature review, citing and integrating recent work effectively, which makes the discussion well situated within ongoing research.

**Weaknesses:**

• The evidence is weak as it is based on a single small case study (rewriting NeurIPS website text with slight modifications), which is not sufficient to support such a strong claim. For example, changes like replacing “features” with “featuring” or “addition” with “complementing” are anecdotal. What if the prompt leads both models to produce similar outputs with the same kinds of supporting words? More systematic experiments or diverse real-world scenarios would make the argument much more convincing.

• By framing detection as simply “impossible” without offering substantial alternatives, is risky. The idea is strong and the role of detectors is important, but a deeper exploration of how to differentiate (and in which contexts it may still be useful) would strengthen the contribution.

• The paper raises valid concerns but does not provide a clear roadmap for the future. Readers may be left uncertain about what practical steps researchers should actually take.

• Counterarguments are not fully engaged with. Some researchers still claim detection is possible under certain conditions, while others disagree.

**Questions:**

- When you describe detection as unsolvable, do you mean it is fundamentally impossible, or simply that it will never be reliable enough for high stakes contexts ?

- This is a very interesting line of work and if refining detectors is not the right path, where should future research focus instead fact checking, provenance tracking, or disclosure standards?

- Do you believe detection will eventually become irrelevant as human and machine writing converge, or is there still potential for developing more effective detection methods?

**Alternative Position:**

Yes, and alternative positions are well-considered and addressed by the argument

**Author Identification:**

No.

**Context:**

4

**Discussion:**

4

**Ethics:**

["NO or VERY MINOR ethics concerns only"]

**Position:**

Yes, the paper argues for or against a position related to machine learning.

**Support:**

3

**Thoroughness:**

3

---

### Note · Authors · 2025-09-02

**1-10 Additional Comments:**

The process is not clear. For example, we don't know if reviewers can see the responses to their questions.

Due to the character limit, we cannot answer all the questions raised by the reviewers.

**1-11 Submit Again:**

Probably yes

**1-1 Submission Process:**

4

**1-2 Next Year:**

Yes, I'd like to.

**1-3 Future Development:**

It might be helpful to set some common topics to encourage everyone to join the conversation.

**1-4 Interest:**

["Panel discussions with other position paper authors", "Structured debates on controversial topics"]

**1-5 Thoughtful:**

8

**1-6 Supportive:**

8

**1-7 Technical Aspects Versus Position:**

5

**1-8 Gate Keeping:**

8

**1-9 Camera Ready Changes:**

We will add more experimental results, including more text samples and more detectors. (Since our initial focus was on presenting our position, we only conducted the case study for illustration.) For example, we use the abstracts of academic papers for simulation, whose detection results also support our position and conclusion.

Based on the reviewers’ feedback, we will also include more discussion and make adjustments to some of the existing sections.

1. Further discussion regarding the future (Reviewer esMk, Krpo). For example, we expect that researchers working on detectors should provide appropriate cautions in their publications. While the hybrid human-in-the-loop model is a possible option, it should be regarded as one way to mitigate the problem, not a complete solution.
2. More detailed overview of detectors and additional references (Reviewer 9JVK). Even though none of the reviewers mentioned the lack of specific references, we will also address their questions by adding relevant citations.
3. Reorganization of repetitive content and the connections between different sections (Reviewer Krpo). For instance, the relevance of certain subsections within Section 5, Issues and Limitations.
4. The correction of possible misunderstandings and supplementary explanations (Reviewer esMk, 9JVK). For example, the detection can only be achieved under multiple assumptions, but in reality, these assumptions are often difficult to satisfy.
5. Other minor suggestions raised by the reviewers.

Due to the character limit, we can’t provide a detailed account here of the changes that will be made in the camera-ready version if the paper is accepted.

We would like to thank the reviewers for their feedback and suggestions. We are grateful that the reviewers mostly agreed with the position of this paper and the way we presented it. We also believe these changes will make our argument more complete and improve the readability of this paper for a general audience.

**3-1 Review Response1:**

esMk

**3-2 Reaction To Review1:**

We thank reviewer esMk for providing a comprehensive summary of the strengths of our paper and for basically agreeing with our position. I think the reviewer's comments are thoughtful and responsible.Their concerns mainly focus on the discussion of future directions, which we will also try our best to address.

We also agree that detectors are important, but their results must be interpreted and applied more rationally.
The detection can only be achieved under multiple assumptions, but in reality, these assumptions are often difficult to satisfy.
We may consider certain texts to be close to the style of a (particular) LLM, but they cannot be used to determine whether they were actually generated by LLMs, especially for people who do not understand how detection tools work.
We believe that while human and machine languages may converge, they will not become entirely identical. Hence, detection will still be useful in the future, even if its role becomes more limited.

**3-3 Review Response2:**

9JVK

**3-4 Reaction To Review2:**

We thank reviewer 9JVK for recognizing the value of our paper and for largely agreeing with our position. I find the reviewer's feedback to be careful and responsible. We will try our best to resolve their questions about the details.

We appreciate the question and suggestions. More examples, more detailed overview and corresponding references will be added.
The prompts are already presented in our paper, within the parentheses following each model name.

**3-5 Review Response3:**

Krpo

**3-6 Reaction To Review3:**

We are grateful to reviewer Krpo for highlighting the strengths of our work and for basically supporting our position. I find the reviewer's comments to be insightful. Since the reviewer is more focused on broader application scenarios, we will add further discussion.

We will revise certain expressions and strengthen the connections between different sections.
We'll add some specific recommendations and practical strategies. While the hybrid human-in-the-loop model is a possible option, it should be regarded as one way to mitigate the problem, not a perfect solution.
As we have written in our paper “Detection efforts should target substantive content (e.g., fact-checking) rather than linguistic characteristics”. Disclosure guidelines may be useful, but their application is limited. Human verification can easily become too costly and therefore not practical for wide adoption.
We agree that the usage should depend on the specific context. For example, different detectors may need to be employed in different cases for assistance. We will elaborate further in the next version.

---

### Meta-Review · Area_Chair_Qqnn · 2025-09-14

**Rating:** 6
**Confidence:** 5

**Strengths:**

The paper takes a clear position, tackling the urgent question of detecting AI-generated text. It highlights ethical dimensions and presents concrete guidelines for good benchmarks while outlining trade-offs and risks in current evaluation pipelines.

**Weaknesses:**

The paper does not clearly discuss alternative positions, offers no concrete solutions, relies on limited evidence from a small case study, and provides insufficient justification for some methodological choices, such as the models analyzed.

**Questions:**

N/A

**Ethics:**

NO or VERY MINOR ethics concerns only

**Thoroughness:**

5

---

### Decision · Program_Chairs · 2025-09-26

Reject